# Pattern, causes and functional outcome of peripheral neuropathies in the Amazon region

Nathalie Deschamps[1,2]*, Mathieu Nacher[2,3], Pierre-Marie Preux[4], Valérie Takam[1], Romain Blaizot[5,6], Beatrice Cenciu[7,8], Nadia Sabbah[9], Bertrand De Toffol[1,2]

1 Department of Neurology, Centre Hospitalier Andree Rosemon, Cayenne, French Guiana, 2 Clinic Investigation Center Antilles Guyane, CIC INSERM1424; Centre Hospitalier Andree Rosemon, Cayenne, French Guiana, 3 Department of Medicine, COREVIH Centre Hospitalier Andree Rosemon, Cayenne, French Guiana, 4 Inserm U1094, IRD U270, EpiMaCT–Epidemiology of Chronic Diseases in Tropical Areas, Institute of Epidemiology and Tropical Neurology, Univ. Limoges, Limoges, France, 5 Department of Dermatology, Centre Hospitalier Andree Rosemon, Cayenne, French Guiana, 6 UMR TBIP Tropical Biomes and Immuno-Physiopathology, University of French Guiana, Cayenne, French Guiana, 7 National Reference Center for Leishmania, Cayenne, French Guiana, 8 Department of Oncology, Centre Hospitalier Andree Rosemon, Cayenne, French Guiana, 9 Department of Endocrinology Diabetology Nutrition, Centre Hospitalier Andree Rosemon, Cayenne, French Guiana

* deschampsnathalie3@gmail.com

**Data Availability Statement:** All relevant data are within the manuscript and its Supporting Information files.

## Abstract

### Background

Peripheral neuropathies have a wide variety of causes and pathophysiologies. Assuming that there are local particularities in the Amazon region, the aim of this study was to describe the characteristics of patients by determining the factors associated with functional severity.

### Methods

This was a retrospective observational study at Cayenne Hospital between January 2015 and May 2023. We included patients from the French hospital activity monitoring database "Programme de médicalisation des systèmes d'information" by searching for the keywords "neuropathy" "Guillain-Barre syndrome"(GBS) "multineuritis" "polyneuritis". The Peripheral Neuropathy Disability (PND) score was determined to quantify the functional severity of patients.

### Results

A total of 754 patients were included, with a mean age of 60 years (SD = 14.6) and a predominance of women (53.6%). Gait disorders were present in 16.3% of patients (PND between 2 and 4). Mortality was 14.3% at 3 years. The most common causes of neuropathy were diabetes (58.2%), chemotherapy induced polyneuropathy toxicity (21.5%), Guillain-Barre syndrome (6.0%), unlabeled (3.2%), and infectious causes (2.0%). Infectious causes mainly included HIV in 13 patients (43.3%) and leprosy in 8 patients (26.7%). Only GBS was significantly associated with functional severity but all patients with a deficiency had a

**Funding:** The author(s) received no specific funding for this work.

**Competing interests:** The authors have declared that no competing interests exist

severe score. But, after excluding GBS, infectious causes were significantly associated with a severe PND score (aOR = 3.69 [1.18–11.58]).

## Discussion

The characteristics of French Guiana combine those found in developed and developing countries, with an over-representation of infections (notably HIV and leprosy), and diabetes. The causes often result from social inequalities in health.

## Conclusion

This is the first comprehensive study of the diverse causes of neuropathy in a territory undergoing epidemiologic transition in the Amazon region.

## Introduction

Peripheral neuropathies (PN) encompass a spectrum of disorders leading to sensory, motor, or autonomic dysfunction. Although their exact incidence and prevalence remain uncertain, a 2015 meta-analysis estimates their prevalence at 1% in the general population and 7% in the elderly, with a greater occurrence in women and in Western countries [1]. PN have diverse etiologies, with approximately 20–30% of cases remaining idiopathic [1]. In Western countries, diabetes constitutes the leading cause, accounting for 18% to 49% of PN cases according to various studies [2–5], followed by alcohol consumption, contributing to 6–10% of cases [2, 6]. By contrast, leprosy is a prevalent cause in developing countries [7, 8], while inflammatory conditions such as Guillain-Barré syndrome (GBS) and chronic inflammatory demyelinating polyneuropathy (CIDP) collectively account for 2–16% of cases [1, 2]. Other less common causes include Hereditary transthyretin amyloidosis (hATTR) responsible of familial amyloid polyneuropathy (FAP) [9] and various genetic disorders [10].

Comparative studies between North and South America have revealed distinct patterns, with South America showing a higher prevalence of diabetes, infections, and FAP [8]. French Guiana, a French territory in Amazonian South America, represents a unique setting characterized by rich biodiversity, including infectious pathogens, the highest Gross Domestic Product (GDP) per capita in Latin America, and significant immigration. The territory is undergoing an epidemiological transition, marked by a dual burden of infectious diseases and chronic non-communicable diseases, both known to be associated with PN. Notably, diabetes prevalence is higher at 9.3% compared to Western countries [11]. Additionally, arboviral diseases are prevalent due to geographical factors [12, 13], alongside a persistently high prevalence of human immunodeficiency virus (HIV) infection exceeding 1% for over three decades [14], and leprosy [15]. However, no previous studies have investigated the prevalence or etiological breakdown of PN in French Guiana or any other Amazonian territory.

Despite variations in their clinical course–acute, sub-acute or chronic–and pathophysiology, PN causes can lead to varying degrees of functional impairment. To our knowledge no studies have yet compared functional severity across different neuropathies. Given French Guiana's unique context with high prevalences of diverse PN causes, a wide array of infectious pathogens, and a dedicated neurology department, our study aims to elucidate the most common PN causes and their associations with functional impairment, particularly gait disturbance.

## Methods

### Design

This study employed a single-center retrospective observational design, focusing on patients who had been hospitalized or consulted at the Cayenne Hospital Center, the sole hospital in French Guiana with a dedicated neurology department, serving as the referral center for all neurological disorders in the region.

### Study participants

Patients were identified from the "Programme de médicalisation des systèmes d'information" database (PMSI), a system tracking inpatient activity in French hospitals, using keywords such as "neuropathy," "Guillain-Barre syndrome," "chronic inflammatory demyelinating polyneuropathy (CIDP)," "multineuritis," "amyloid," and "amyloidosis." Inclusion criteria comprised patients with neuropathy onset between January 2015 and May 2023, while exclusion criteria encompassed tunnel syndrome, radiculopathies, pre-existing symptoms predating January 2015, and insufficient medical information regarding neuropathy characteristics to calculate the Peripheral Neuropathy Disability (PND) functional scores. Patients were informed of the study by mail to obtain their non-opposition, as required by the law.

### Definition of neuropathy

Peripheral neuropathy was defined as a sensory and/or motor deficit associated with abolition of reflexes, with or without electroneuromyography confirmation after evaluation by a neurologist, except for patients with diabetes and cancer, which were diagnosed by an endocrinologist and an oncologist, respectively.

Diabetic neuropathy diagnoses were made by endocrinologists, grading diabetic foot using International Working Group on the Diabetic Foot guidelines [16], categorized as "diabetic." Chemotherapy-related toxicity diagnoses were conducted by oncologists, utilizing the Total Neuropathy Score (TNS) [17], categorized as "toxic."

Neuropathic pain was defined as a painful sensation of burning, tingling, numbness, itching, electric discharge or cold.

### Determining the cause of the neuropathy

Causes were recorded in medical records and classified according to etiology, including "Guillain-Barre syndrome" (GBS), "alcoholic," "deficiency," "infectious," "chronic renal failure (CRF)," "connective tissue disease (CTD)," "vasculitis," "rheumatism," "hereditary pressure hypersensitivity neuropathy (HNPP)," "CIDP," "FAP," "monoclonal gammopathy," "toxic," "critical illness polyneuropathy (CIP)" or "unlabeled." Patients were classified as "unlabeled" if the results of etiological screening were negative or not available. Also, some patients with a suspected genetic cause or incomplete etiologic assessment were classified as unlabeled because they had interrupted follow-up. Guillain-Barré syndrome diagnoses followed Brighton criteria [18], while CIDP diagnoses adhered to EAN/PNS criteria [19]. Infectious neuropathies were subtyped as "Leprosy," "Human immunodeficiency virus (HIV)," "ZIKA,"(ZIKV) "Dengue,"(DENV) "Chikungunya,"(CHIK) "CMV," "Campylobacter jejuni"or "SARS-CoV2."

### Collected data

Clinical data supporting neuropathy diagnosis, including paresthesia, sensorimotor deficits, and reflex abolition, were extracted from medical records to confirm peripheral neuropathy

and to determine PND score. Variables included age at symptom onset, year of symptom onset, year of last follow-up, and mortality.

## Outcome measures

The primary endpoint was the Peripheral Neuropathy Disability (PND) functional scores which were retrospectively determined from medical records to quantify functional severity at disease onset, 1 year and 2 years. PND score is defined as follows: "1":sensory symptoms, walking normally; "2":walking difficulty does not require support or stick; "3"One or two sticks or crutches needed for deambulation; "4": patient confined in bed or wheelchair.

The secondary endpoints were the presence of neuropathic pain at baseline, the mortality at 1 year, 2 years, 3 years.

## Statistical analysis

Patient characteristics and clinical data were recorded in an anonymized Excel file. Subsequently, data were extracted and analyzed utilizing STATA 18 software (StataCorp, College Station, Texas, USA). Age was presented as mean ± standard deviation (SD). A category of age was defined as under 50 years old (age<50yo) and reported as numbers and percentages like while sex, mortality, and causes.

To compare functional severity between groups, patients were divided into two categories: "severe PND" for scores between 2 and 4, indicative of walking problems, and "non-severe PND" for a score of 1. The two groups were compared using a t-test for age and Chi-squared test to compute odds ratios (OR) for sex, death, and causes.

Assessment of functional severity and neuropathic pain within each group was conducted through cross-tabulation and Chi-squared tests. Subsequently, multiple logistic regression was employed to calculate adjusted odds ratios (aOR) for age and sex. To evaluate the evolution of PND score by cause, mean PND at 1 year and 2 years were compared to the mean PND score at baseline using a t-test. To study mortality by cause, we estimated incidence rates per 1000 person-years. Survival analysis was also performed with death as failure variable; after the date of last visit patients were censored; this allowed to plot Kaplan-Meier survivor curves.

## Ethical and regulatory aspects

The study is classified under the MR-004 reference methodology for research in the public interest of the Commission Nationale de l'Informatique et des Libertés (CNIL) (Deliberation n ˚ 2018–155 of May 3, 2018) for which the Cayenne Hospital Center signed a compliance commitment on 26/09/2023. In accordance with the European General Data Protection Regulation, a Data Protection Impact Assessment (DPIA) was carried out and validated by the Data Protection Officer (DPO) of the Cayenne Hospital Center on 26/09/2023. In addition, this study was entered in the hospital's internal treatment registry by the DPO and a summary of this study has been published on the Health-Data-Hub (HDH) website. The information note was sent to the patient/participant's mailing address. If the patient/participant did not express any objection within one month from the date the letter was sent, it was considered that he or she did not object to participating in the study. In accordance with the legislation for this type of research, all the mandatory formalities have been completed. Under no circumstances is the opinion of an ethics committee (committee for the protection of persons, local ethics committee or ethics committee with an ID-RCB number) mandatory for this type of research according to the regulations in force in France.

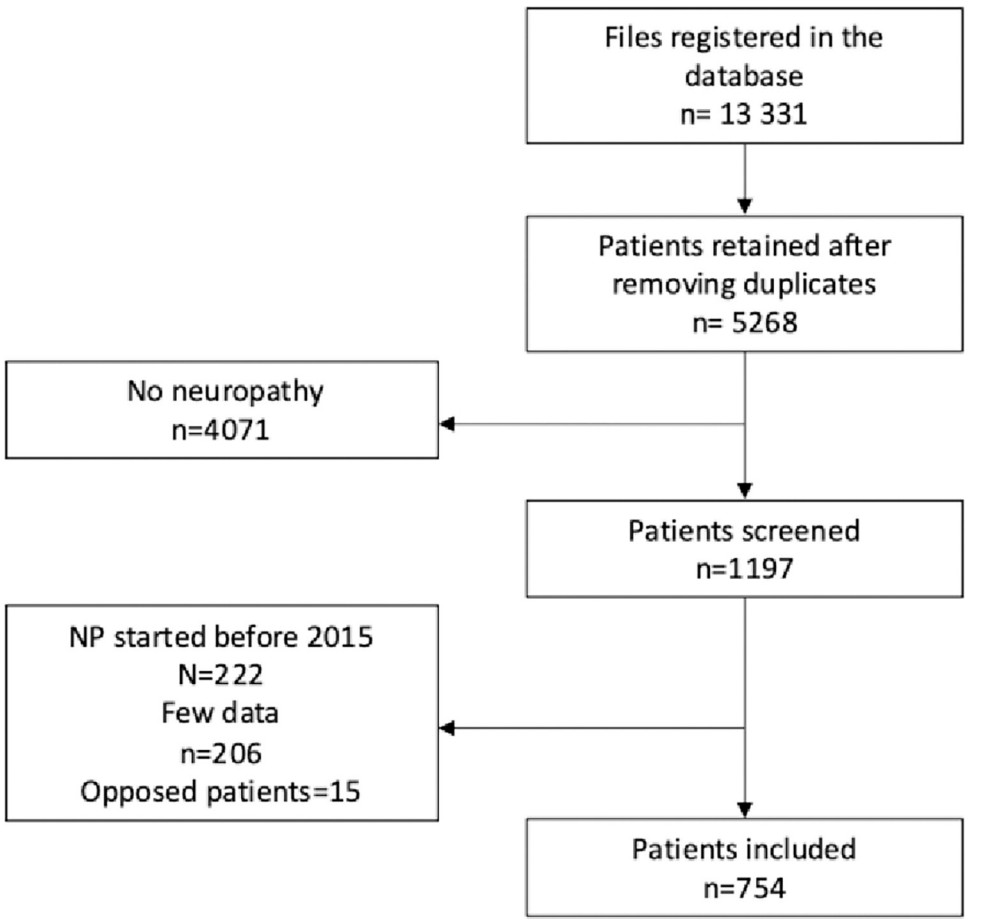

**Fig 1. Flow chart of the 754 patients included.** Notes: Some patients had several files in the database.

## Results

### Characteristics of patients

Overall, 754 patients were included in the study (Fig 1). Among them, 125 (16.6%) had an electroneuromyogram (ENMG). The mean age was 60 years (SD = 14.6), with women being slightly over-represented (53.6%) (Table 1). Among the patients, 631 (83.7%) had a PND score of 1, 60 (8.0%) had a score of 2, 19 (2.5%) had a score of 3, and 44 (5.8%) had a score of 4 (Fig 2). We had data to estimate the PND score for 406 patients at 1 year, and 217 patients at 2 years.

The most frequent causes, in descending order, were diabetes, toxicity (98.8% related to neurotoxic chemotherapy), GBS, multiple causes, unlabeled causes, and infections. Among, GBS, 9(16.4%) had an associated infection with a positive virology test for SARS-CoV2 for 2 patients, CMV for 1, ZIKV for 2, DENV for 1, CHIK for 1, *Campylobacter jejuni* for 2. Infectious causes were mainly HIV in 13 patients (43.3%) and leprosy in 8 patients (26.7%) (Fig 3). Among toxic causes, 1 (0.6%) patient consumed an herbal concoction "Bita," 1 (0.6%) was exposed to mercury and others (98.8%) overwhelmingly had chemotherapy-induced peripheral neuropathy (CIPN). The chemotherapies were platin-based for 87 patients (54.4%), taxol-based for 68 patients (42.5%), vincristine for 3 patients (1.9%) and other molecules for 35 patients (21.9%). Deficiency-related causes were vitamin-B12 deficiency for 3 patients (60%), vitamin B9 deficiency for 1 patient (20%) and vitamin E deficiency for 1 patient (20%).

**Table 1. Characteristics of the 754 patients included with peripheral neuropathy.**

| | PND score | | | Crude OR | Adjusted OR |
|---|---|---|---|---|---|
| | **1** | **2–4** | **Total** | **[95%CI]** | **[95%CI]** |
| | **N = 631 (83.7%)** | **N = 123 (16.3%)** | **N = 754 (100.0%)** | | |
| **Age mean(SD)** | 61.1 (12.5) | 54.3 (21.5) | 60.02 (14.6) | | |
| Age<50 yo n(%) | 112 (17.7) | 45(36.6) | 157 (20.8) | 2.7 [1.7–4.1] | 1.02 [0.6–1.7] |
| **Sex** | | | | | |
| Female n(%) | 351 (55.7) | 53 (42.7) | 404 (53.6) | | |
| Male n(%) | 279 (44.2) | 71 (57.7) | 350 (46.4) | 1.7 [1.1–2.6] | 0.97 [0.6–1.5] |
| **Causes** | | | | | |
| Alcohol n(%) | 9 (1.4) | 4 (3.3) | 13 (1.7) | 2.3 [0.5–8.5] | |
| Diabetes n(%) | 412 (65.3) | 27 (22) | 439 (58.2) | 0.1 [0.1–0.2] | 0.2 [0.1–0.3] |
| Deficiency n(%) | 0 (0) | 5 (4.1) | 5 (0.7) | - | |
| CRF n(%) | 0 (0) | 1 (0.8) | 1 (0.1) | - | |
| FAP n(%) | 2 (0.3) | 2 (1.6) | 4 (0.5) | 5.2 [0.4–72.1] | |
| HNPP n(%) | 0 (0) | 1(0.8) | 1 (0.1) | - | |
| Infections n(%) | 10 (1.6) | 5(4.1) | 15 (2.0) | 2.6 [0.7–8.6] | |
| CIDP n(%) | 0(0) | 2(1.6) | 2 (0.3) | - | |
| Vasculitis n(%) | 6 (1) | 2(1.6) | 8 (1.1) | 1.7 [0.2–9.8] | |
| CTD n(%) | 3 (0.5) | 1(0.8) | 4 (0.5) | 1.7 [0.03–21.6] | |
| MG n(%) | 1 (0.2) | 2(1.6) | 3 (0.4) | 10.4 [0.5–615.2] | |
| Toxicity n(%) | 156 (24.7) | 6(4.9) | 162 (21.5) | 0.2 [0.1–0.4] | 0.16 [0.1–0.4] |
| Rheumatism n(%) | 1 (0.2) | 1 (0.8) | 2 (0.3) | 5.2 [0.1-406] | |
| CIP n(%) | 0 (0) | 2 (1.6) | 2 (0.3) | - | |
| Unlabeled n(%) | 18 (2.9) | 6 (4.9) | 24 (3.2) | 1.7 [0.6–4.7] | |
| GBS n(%) | 1(0.2) | 44 (35.8) | 45 (6.0) | 350.89[57.39–14187.97] | 306.23[41.42–2263.76] |
| Multiple causes n(%) | 12(1.9) | 12 (9.8) | 24 (3.2) | 5.58[2.22–13.91] | 5.37[2.29–12.62] |

Abbreviations: CIDP: chronic inflammatory demyelinating polyneuropathy; CIP = critical illness polyneuropathy; CRF = chronic renal failure; CTD: connective tissues disease; FAP: familial amyloid polyneuropathy; GBS = Guillain-Barré syndrome; HNPP = Hereditary neuropathy with liability to pressure palsies; MG = monoclonal gammopathy; OR = odd ratio; PND = Peripheral Nerve Disability

Statistical analysis: Chi-squared test for crude OR; logistic regression for aOR

## Primary endpoint

Among all causes, the only factor significantly associated at baseline with functional severity was GBS but all patients with a deficiency-related NP had a severe score. Diabetic and toxic causes were significantly less severe (Table 1). However, after excluding GBS, which is a known severe neuropathy, infectious causes were significantly associated with severe PND score (aOR = 3.69 [1.18–11.58]) (Fig 4). At 1 year, GBS and deficiency-related NP were significantly associated with a severe PND score whereas only toxic cause remained significantly less severe (Table 2). Diabetes presented a functional worsening with a significant increase of the PND score at 1 and 2 years. Conversely, toxic neuropathy and Guillain-Barré syndrome improved their functional scores at 1 and 2 years (Fig 5).

## Secondary outcome

Neuropathic pain was present in 18.2% of cases. The causes associated with presence of neuropathic pain were alcohol, FAP, vasculitis, unlabeled causes, GBS and multiple causes (Table 3).

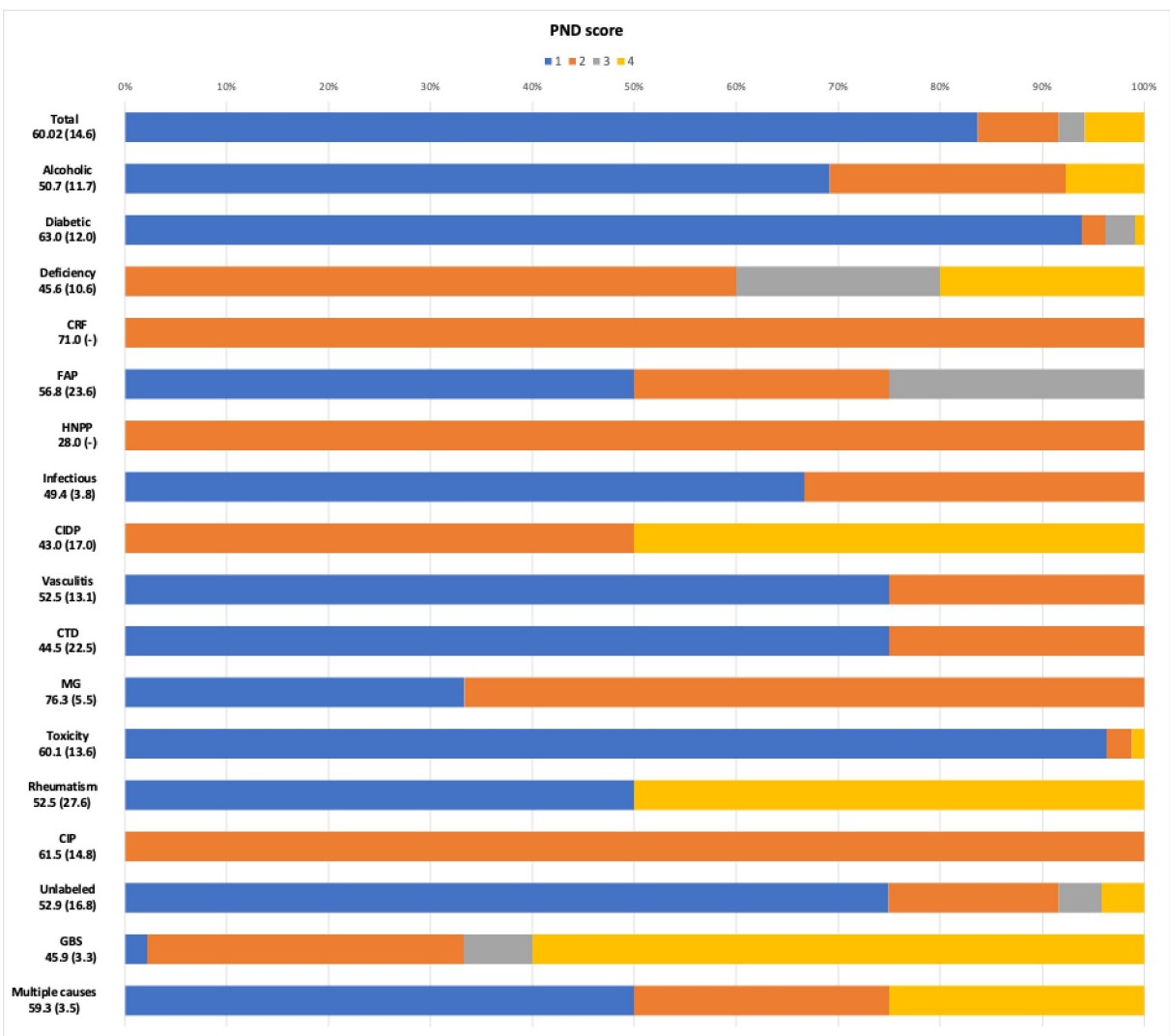

**Fig 2. Repartition of PND score by causes of peripheral neuropathy.** Cause Age years old mean(SD); PND score 1/2/3/4 (%): Total = 83.7/ 8.0/2.5/5.8; Alcoholic = 69.2/23.1/0.0/7.7; Diabetic = 93.8/2.3/2.9/0.9; Deficiency = 0.0/60.0/20.0/20.0; CRF (Chronic renal failure) = 0.0/100.0/ 0.0/0.0; FAP (Familial amyloid Polyneuropathy) = 50.0/25.0/25.0/0.0; HNPP(Hereditary neuropathy with liability to pressure palsies) = 0.0/ 100.0/0.0/0.0; Infectious = 66.7/33.3/0.0/0.0; CIDP (chronic inflammatory demyelinating polyneuropathy) = 0.0/50.0/0.0/50.0; Vasculitis = 75.0/25.0/0.0/0.0; CTD (connective tissues disease) = 75.0/25.0/0.0/0.0; MG (Monoclonal gammopathy) = 33.3/66.6/0.0/0.0; Toxicity = 96.3/2.5/0.0/1.2; Rheumatism = 50.0/0.0/0.0/50.0; CIP (Critical illness polyneuropathy) = 0.0/100.0/0.0/0.0; Unlabeled = 0.0/100.0/ 0.0/0.0; GBS(Guillain-Barré syndrome) = 2.2/31.1/6.7/60.0; Multiple causes = 50.0/25.0/0.0/25.0.

Mortality was 4.1% at 1 year, 9.5% at 2 years and 14.3% at 3 years (Table 4). Deceased patients at 3 years were mainly those with diabetic neuropathy (32.5%) or with toxic neuropathy (47.5%) but the probability of death also increased for infectious, unlabeled causes and multiples causes of neuropathy (Fig 6).

## Discussion

Our study reveals both similarities and differences with the existing literature. Consistent with findings from high-income countries, peripheral neuropathy (PN) predominantly affected women [20], individuals over 50 years old, and exhibited a notable association with diabetes,

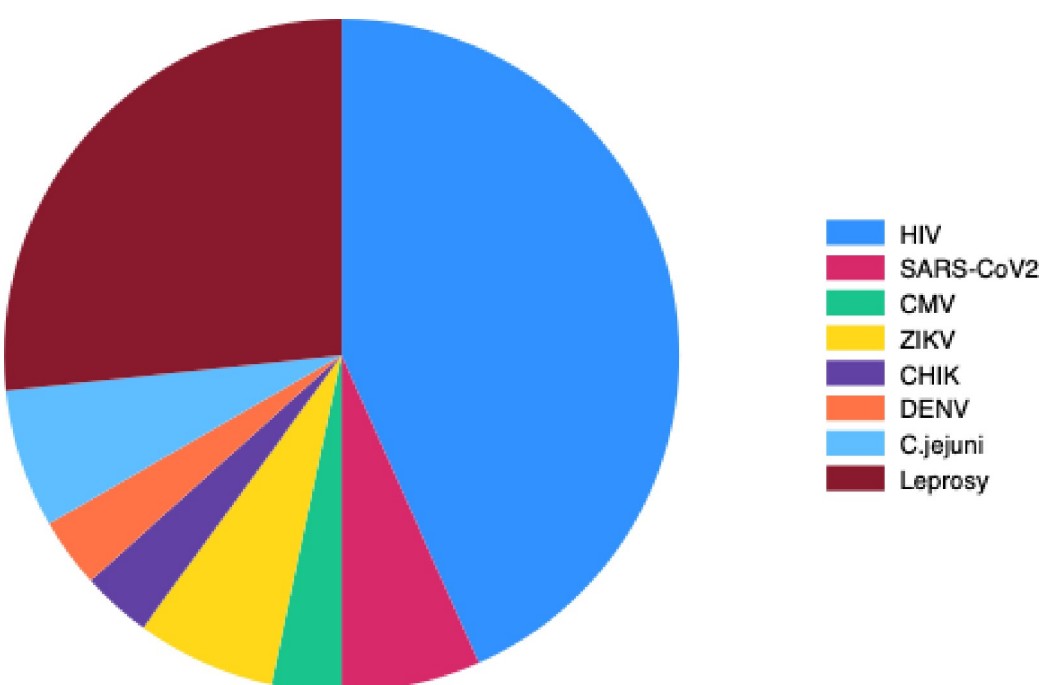

**Fig 3. Repartition of infectious causes of peripheral neuropathy.** Repartition by infectious cause (n(%)): Campylobacter jejuni 2(6.7); CHIK 1 (3.3%) CMV 1(3.3); DENV 1 (3.3); HIV 13(43.3); Leprosy 8(26.7); SARS-Cov2 2(7.4); ZIKV 2(6.7). Note: patients with multiple causes of neuropathy were included.

as reported elsewhere [1]. However, French Guiana presents a unique amalgamation of features characteristic of both Western and developing countries. Notably, there was a notable over-representation of infectious causes, particularly leprosy, akin to patterns observed in

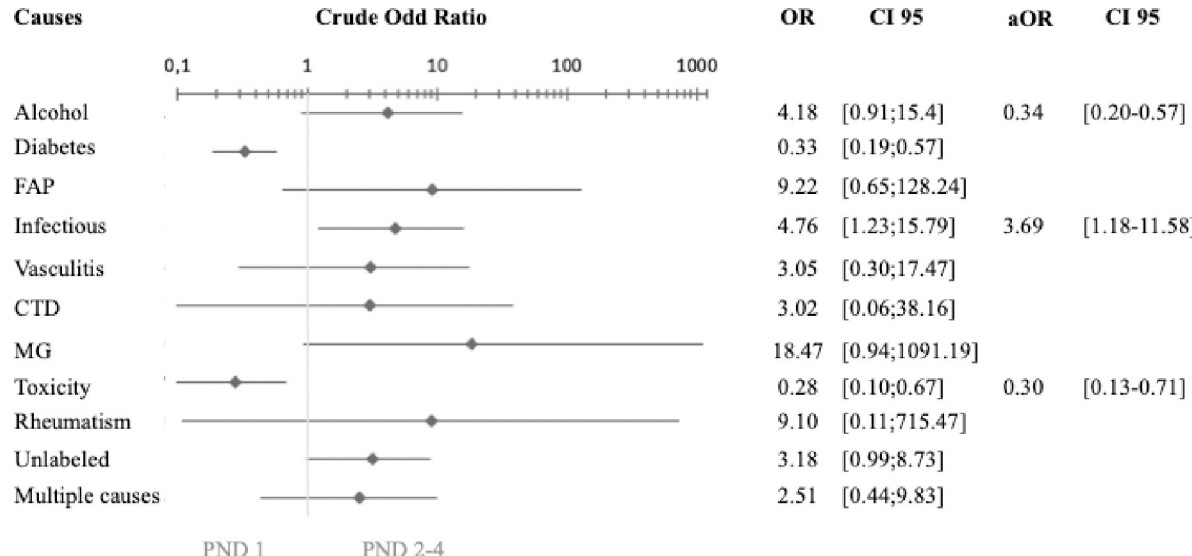

**Fig 4. Association between causes of peripheral neuropathy and severe PND score excluding GBS.** Abbreviations: aOR = adjusted odd ratio; CI = confidence interval; HNPP = Hereditary neuropathy with liability to pressure palsies; CTD: connective tissues disease; FAP: familial amyloid polyneuropathy; MG = monoclonal gammopathy; OR = odd ratio; PND = Peripheral Nerve Disability. Statistical analysis: Chi-squared test for crude OR; logistic regression for aOR.

**Table 2. Risk factors associated with severe PND at 1 year.**

| | Severe PND at 1 year (N = 406) | |
|---|---|---|
| | **OR [CI 95]** | **aOR[CI 95]** |
| Alcohol | - | |
| Diabetes | 0.59 [0.34–1.02] | |
| Deficiency | 14.74 [1.15–775.66] | 14.06 [1.43–138.47] |
| CRF | - | |
| FAP | 9.68 [0.49–572.73] | |
| HNPP | - | |
| Infectious | 1.59 [0.15–9.12] | |
| CIDP | - | |
| Vasculitis | 2.40 [0.21–17.06] | |
| CTD | - | |
| MG | - | |
| Toxicity | 0.41 [0.18–0.85] | 0.39 [0.19–0.80] |
| Rheumatism | 4.77 [0.06–375.61] | |
| CIP | - | |
| Unlabeled | 1.76 [0.40–6.16] | |
| GBS | 33.86 [7.17–314.86] | 34.87 [7.46–163.06] |
| Multiple causes | 1.05 [0.11–5.23] | |

Abbreviations: CIP = critical illness polyneuropathy; CRF = chronic renal failure; HNPP = Hereditary neuropathy with liability to pressure palsies; CIDP: chronic inflammatory demyelinating polyneuropathy; CTD: connective tissues disease; FAP: familial amyloid polyneuropathy; GBS = Guillain-Barré syndrome MG = monoclonal gammopathy; OR = odd ratio

Statistical analysis: Chi-squared test for crude OR; logistic regression for aOR

developing nations. By contrast, causes such as diabetes, CIPN, and alcohol were more prevalent than in developing countries, reminiscent of trends in Western countries (Table 1).

This over-representation of infectious causes can be attributed to the high incidence of leprosy and HIV in French Guiana (Fig 3). Despite a previous decline in the incidence of leprosy in French Guiana since the 1980s, a resurgence of cases has been observed since 2006, with an incidence of > 1 case per 10,000 inhabitants exceeding the threshold set by the World Health Organization (WHO) [15]. Notably, a significant proportion of these cases involve Brazilian immigrants, many of whom immigrate to French Guiana, and for whom leprosy neuropathies have been extensively documented [21]. French Guiana also exhibits the highest prevalence of HIV infection among all French territories, surpassing 1% [14]. These are mainly immigrants who often contracted the virus in French Guiana [22]. Unfortunately, diagnosis was made at a late stage in 24 to 35% of cases [23], likely contributing to the over-representation of peripheral neuropathy (PN) in our population. Notably, HIV-infected patients who develop neuropathy tend to have lower CD4 count [24], indicative of advanced disease progression.

Diabetic patients were over-represented—echoing the high prevalence of diabetes in French Guiana [25]—in association with patients who were obese (18.8%) or overweight (35.9%), and often had a low social status [11]. Furthermore, the nutritional balance is frequently suboptimal [26, 27] with undernutrition notably contributing to severe neuropathies (Fig 2). These findings underscore the very high levels of social inequality in French Guiana, characterized by a high rate of precariousness, which poses significant challenges to the effective management of metabolic diseases [28].

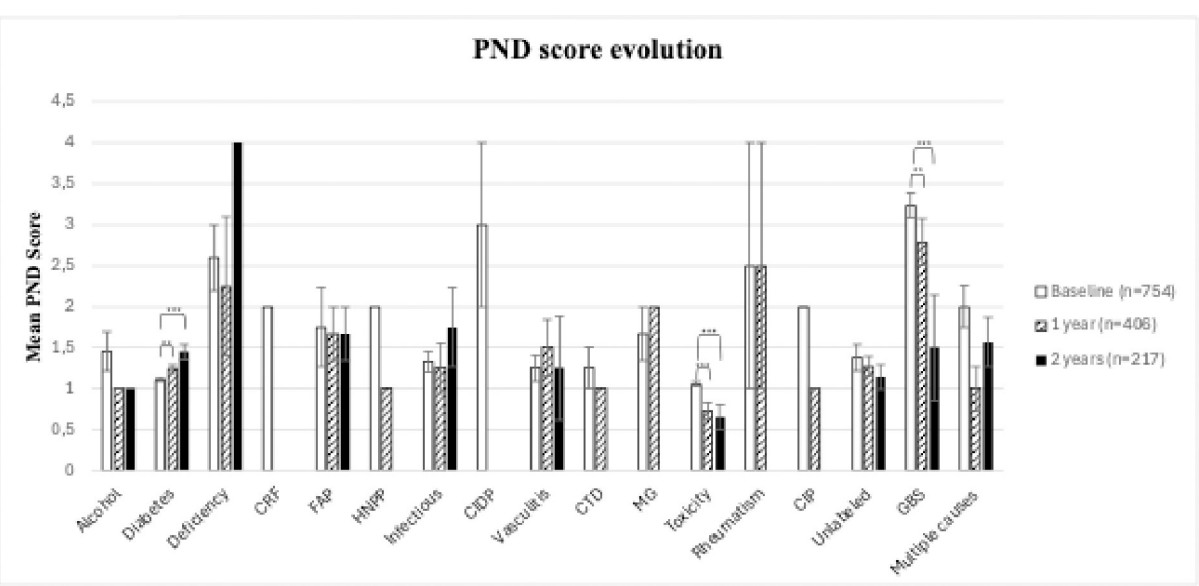

**Fig 5. Mean PND score with standard deviation by causes of peripheral neuropathy at baseline, 1 year and 2 years.** ** <0.01; *** <0.001:
CRF = chronic renal failure; CIDP = chronic inflammatory demyelinating polyneuropathy; CIP = critical illness polyneuropathy;
CTD = connective tissues disease GBS = Guillain-Barré syndrome; FAP: familial amyloid polyneuropathy; HNPP = Hereditary neuropathy
with liability to pressure palsies; MG = monoclonal gammopathy; PND = Peripheral Nerve Disability. Mean (SD) PND score at baseline:
Alcohol = 1.5(0.3);Diabetes = 1.1(0.0); Deficiency = 2.6(0.4); CRF = 2.0(-); FAP = 1.8(0.5); HNPP = 2.0(-); Infectious = 1.3(0.1); CIDP = 3(1);
Vasculitis = 1.3(0.2); CTD = 1.3(0.3);MG = 1.7(0.3);Toxicity = 1.1(0.0); Rheumatism = 2.5(1.5); CIP = 2.0(0.0); Unlabeled = 1.4(0.2); GBS = 3.2
(0.2); Multiple causes = 2(0.3). Mean (SD) PND score at 1 year: Alcohol = 1.0(0.0); Diabetes = 1.2(0.1); Deficiency = 2.3(0.1); FAP = 1.7(0.3);
HNPP = 1.0 (-); Infectious = 1.3(0.3);Vasculitis = 1.5(0.3); CTD = 1.0(-); MG = 2.0(-);Toxicity = 0.7(0.1); Rheumatism = 2.5(1.5); CIP = 1.0(0);
Unlabeled = 1.3(0.1); GBS = 2.8(0.3); Multiple causes = 1.0(0.3). Mean (SD) PND score at 2 years: Alcohol = 1.0(0); Diabetes = 1.5(0.1);
Deficiency = 4(-); FAP = 1.7(0.3; Infectious = 1.8 (0.5); Vasculitis = 1.3(0.6); Toxicity = 0.7(0.2); Unlabeled = 1.1(0.2); GBS = 1.5(0.7); Multiple
causes = 1.6(0.3). Statistical analysis: t-test to compare mean PND at 1year and 2 years versus baseline.

A recent study [29] indicates a worldwide prevalence of FAP ranging from 5,500 to 38,500
individuals for a global population of approximately 7.9 billion. Even at the upper end of these
estimates, our findings for French Guiana (which may have missed some cases outside Cay-
enne) suggest a significant over-representation of this neuropathy. We identified 4 patients in
a population of around 300,000, which exceeds the upper end of global estimates by nearly
3000 times.

Our study shows that in our region, when faced with severe PN, the first cause to be consid-
ered is Guillain-Barré syndrome. However, once this has been ruled out, an infectious cause
should be sought, particularly HIV infection or leprosy (Fig 4). In addition, any neuropathy
associated with neuropathic pain should be investigated for alcoholism, vasculitis or amyloid-
osis (Table 3). Despite treatment, patients with GBS or deficiency neuropathy remain signifi-
cantly functionally more severe at 1 year (Table 2).

Although diabetic patients had a significantly lower PND score, there was a progressive
worsening of function at 1 year and 2 years (Fig 5), perhaps highlighting the lack of manage-
ment of this PN, even though it is detected early (as part of the follow-up assessment of dia-
betic patients).

We observed an increased probability of death in patients with diabetic, toxic, infectious
and unlabeled neuropathies (Fig 6).

It is important to acknowledge the limitations of our retrospective, hospital-based study.
The diagnosis of diabetic and CIPN benefits from a screening to prevent their occurrence and
probably explains the over-representation of these causes in our hospital cohort and the lower

**Table 3. Causes associated with neuropathic pain.**

| | No Pain | Pain | OR [CI 95] | aOR [CI 95] |
|---|---|---|---|---|
| | N = 617 (81.8%) | N = 137 (18.2%) | | |
| Alcohol n(%) | 7 (53.8) | 6 (46.2) | 3.99 [1.09–14.09] | 3.77 [1.24–11.49] |
| Diabetes n(%) | 365 (83.3) | 73 (16.7) | 0.87 [0.58–1.31] | |
| Deficiency n(%) | 3 (60.0) | 2 (40.0) | 3.18 [0.26–27.99] | |
| CRF n(%) | 1 (100.0) | 0 (0.0) | - | |
| FAP n(%) | 1 (25.0) | 3 (75.0) | 14.47 [1.15–760.83] | 12.73 [1.30–124.23] |
| HNPP n(%) | 1 (100.0) | 0 (0.0) | - | |
| Infectious n(%) | 12 (80.0) | 3 (20.0) | 1.18 [0.21–4.48] | |
| CIDP n(%) | 2 (100.0) | 0 (0.0) | - | |
| Vasculitis n(%) | 3 (37.5) | 5 (62.5) | 8.14 [1.55–52.89] | 7.41 [1.74–31.55] |
| CTD n(%) | 3 (75.0) | 1 (25.0) | 1.58 [0.03–19.81] | |
| MG n(%) | 3 (100.0) | 0 (0.0) | - | |
| Toxicity n(%) | 152 (93.8) | 10 (6.2) | 0.25 [0.11–0.50] | 0.24 [0.12–0.47] |
| Rheumatism n(%) | 2 (100.0) | 0 (0.0) | - | |
| CIP n(%) | 2 (100.0) | 0 (0.0) | - | |
| Unlabeled n(%) | 14 (58.3) | 10 (41.7) | 3.57 [1.38–8.86] | 3.26 [1.41–7.54] |
| GBS n(%) | 31 (68.8) | 14 (31.1) | 2.15[1.02–4.31] | 1.99[1.01–3.91] |
| Multiple causes n(%) | 15 (62.5) | 9 (37.5) | 2.82 [1.06–7.04] | 2.82 [1.20–6.64] |

Abbreviations: aOR = adjusted odd ratio; CI = confidence interval; CIP = critical illness polyneuropathy; CRF = chronic renal failure; HNPP = Hereditary neuropathy with liability to pressure palsies; GBS = Guillain-Barré syndrome; CIDP: chronic inflammatory demyelinating polyneuropathy; CTD: connective tissues disease; FAP: familial amyloid polyneuropathy; MG = monoclonal gammopathy; OR = odd ratio; PND = Peripheral Nerve Disability

Statistical analysis: logistic regression for aOR

PND score. Although the ENMG is a confirmatory test for the vast majority of PN, used as an argument in support, it can only confirm damage to large nerve fibers and its normality cannot exclude a small fibers PN. However, the fact that it is only performed on 125 patients could lead to a risk of misdiagnosis in patients with suspected central nervous system involvement as differential diagnosis. We may have missed cases from more isolated regions where access to healthcare is limited, potentially leading to an underrepresentation of some causes of peripheral neuropathies in our findings. Additionally, the retrospective determination of functional scores may have underestimated the severity if documentation of walking difficulties or the need for technical assistance was lacking in patient files.

Furthermore, due to the retrospective nature of this study, physical examination was not systematically performed and was thus not exhaustive, making it impossible to use other scores to characterize PN. In the case of neuropathic pain, there was probably a measurement bias due to the absence of a screening questionnaire which may have led to underestimation. The use of keywords as "Guillain-Barré syndrome", "Chronic inflammatory demyelinating polyneuropathy (CIDP), "amyloid" and "amyloidosis" may have led to a selection bias, with an over-representation of patients with these causes of neuropathy. patients with a negative etiological assessment were also classified as unlabeled

Also, some patients with a suspected genetic cause—linked to discordance between clinical and electrical analysis–or incomplete etiologic assessment were classified as unlabeled because they had interrupted follow-up. Finally, in the local context of widespread poverty, transport difficulties, geographical isolation, intense migration within South America, and low health professional density, irregular follow-up is a major challenge for all health problems in French

**Table 4. Mortality rate and rate of mortality by cause of peripheral neuropathy.**

| | Mortality | | | Mortality rate | |
|---|---|---|---|---|---|
| | **1 year** | **2 years** | **3 years** | **Per 1000 py** | **CI 95** |
| | **N = 568** | **N = 380** | **N = 279** | | |
| Total n(%) | 23 (4.1) | 36 (9.5) | 40 (14.3) | 32.24 | 24.36–42.65 |
| Alcohol n(%) | 0(0) | 0(0) | 0(0) | 0 | - |
| Diabetes n(%) | 6(26.1) | 12(33.3) | 13(32.5) | 25.96 | 17.54–38.42 |
| Deficiency n(%) | 0(0) | 0(0) | 0(0) | 0 | |
| CRF n(%) | 0(0) | 0(0) | 0(0) | 0 | |
| FAP n(%) | 1(4.3) | 1(2.8) | 1(2.5) | 166.67 | 23.48–1183.18 |
| HNPP n(%) | 0(0) | - | - | 0 | |
| Infections n(%) | 1(4.3) | 1(2.8) | 1(2.5) | 23.26 | 3.28–165.09 |
| CIDP n(%) | - | - | - | - | - |
| Vasculitis n(%) | 0(0) | 0(0) | 0(0) | 0 | - |
| CTD n(%) | 0(0) | 0(0) | 0(0) | 125 | 17.61–887.38 |
| MG n(%) | 0(0) | 0(0) | 0(0) | 0 | - |
| Toxicity n(%) | 10(43.5) | 17(47.2) | 19(47.5) | 63.60 | 40.07–100.95 |
| Rheumatism n(%) | 1(4.3) | 1(2.8) | 1(2.5) | 500.00 | 70.43–3549.54 |
| CIP n(%) | 0(0) | - | - | 0 | - |
| Unlabeled n(%) | 2(8.6) | 2(5.6) | 3(7.5) | 23.81 | 3.35–169.03 |
| GBS n(%) | 0(0) | 0(0) | 0(0) | 0 | - |
| Multiple causes n(%) | 2(8.6) | 2(5.6) | 2(5) | 15.15 | 2.13–107.56 |

Abbreviations: CIDP: chronic inflammatory demyelinating polyneuropathy; CIP = critical illness polyneuropathy; CRF = chronic renal failure; CTD: connective tissues disease; FAP: familial amyloid polyneuropathy; GBS = Guillain-Barré syndrome; HNPP = Hereditary neuropathy with liability to pressure palsies; MG = monoclonal gammopathy; OR = odd ratio; PND = Peripheral Nerve Disability; py = person-years.

Guiana, which makes the ascertainment of the vital status at a given time uncertain, until they eventually come back. We therefore censored data after the last known status.

Despite these limitations the present study provides a first estimate of PN in an Amazonian context and offers perspective for future research.

Further studies are thus warranted to elucidate the impact of various exposures on peripheral neuropathies, including factors such as dietary habits (associated with ethnic and cultural differences), plant consumption such as Bita concoctions (which has been implicated in approximately forty cases of Guillain-Barré syndrome [30]), exposition to mercury [31], nutritional status, and social inequalities—characteristic of our region. These investigations are crucial for understanding and clarifying the 3.2% of cases with unexplained causes (Table 1).

## Conclusion

The present study represents the first comprehensive description of peripheral neuropathy (PN) causes in an Amazonian region. The diverse array of PN causes underscores French Guiana's status as a territory straddling the line between developed and developing countries. This study underscores the imperative to implement preventative measures, especially considering French Guiana's well-funded health system. Our findings indicate that the study population faced heightened exposure to diabetes and infections and developed severe neuropathies often associated with undernutrition, all of which are preventable risk factors.

**Fig 6. Risk of death according to causes of peripheral neuropathy.** Kaplan-Meier curve showing probability of death according to causes. Follow-up data was not available for 198 patients. 556 remaining observations are shown. Plot shows number censored at each censoring time. Others group brings together alcoholic neuropathy, Deficiency-related neuropathy, neuropathy with chronic renal failure, critical illness polyneuropathy, Hereditary neuropathy with liability to pressure palsies, Guillain-Barré syndrome, neuropathy with connective tissues disease, familial amyloid polyneuropathy, neuropathy with monoclonal gammopathy. Table with number at risk and number censored is available in S1 Table.

## Supporting information

**S1 Table. Number at risk and number censored in Kaplan-Meir survey analysis.**
(DOCX)

## Acknowledgments

We thank Professor Jean Michel VALLAT from the Department of Neurology at the University Hospital of Limoges and Fabrice QUET from the Clinic Investigation Center Antilles Guyane for their collaborations.

## Author Contributions

**Conceptualization:** Nathalie Deschamps.

**Formal analysis:** Nathalie Deschamps.

**Writing – original draft:** Nathalie Deschamps.

**Writing – review & editing:** Mathieu Nacher, Pierre-Marie Preux, Valérie Takam, Romain Blaizot, Beatrice Cenciu, Nadia Sabbah, Bertrand De Toffol.

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
