## [Decision Letter · Decision Letter 0]

25 Sep 2024

PONE-D-24-27124Severity of peripheral neuropathies among causes in Amazon regionPLOS ONE

Dear Dr. DESCHAMPS,

Thank you for submitting your manuscript to PLOS ONE. After careful consideration, we feel that it has merit but does not fully meet PLOS ONE’s publication criteria as it currently stands. Therefore, we invite you to submit a revised version of the manuscript that addresses the points raised during the review process.

Please submit your revised manuscript by Nov 09 2024 11:59PM. If you will need more time than this to complete your revisions, please reply to this message or contact the journal office at plosone@plos.org. Please include the following items when submitting your revised manuscript:A rebuttal letter that responds to each point raised by the academic editor and reviewer(s). You should upload this letter as a separate file labeled 'Response to Reviewers'.A marked-up copy of your manuscript that highlights changes made to the original version. You should upload this as a separate file labeled 'Revised Manuscript with Track Changes'.An unmarked version of your revised paper without tracked changes. You should upload this as a separate file labeled 'Manuscript'.

We look forward to receiving your revised manuscript.

Kind regards,

Claudia Brogna

Academic Editor

PLOS ONE

**Journal Requirements:**

2. We note that your Data Availability Statement is currently as follows: All relevant data are within the manuscript and its Supporting Information files

Reviewers' comments:

Reviewer's Responses to Questions

**Comments to the Author**

1. Is the manuscript technically sound, and do the data support the conclusions?

Reviewer #1: Yes

Reviewer #2: Yes

Reviewer #3: Partly

Reviewer #4: Partly

2. Has the statistical analysis been performed appropriately and rigorously? 

Reviewer #1: Yes

Reviewer #2: Yes

Reviewer #3: Yes

Reviewer #4: Yes

3. Have the authors made all data underlying the findings in their manuscript fully available?

Reviewer #1: Yes

Reviewer #2: Yes

Reviewer #3: Yes

Reviewer #4: No

4. Is the manuscript presented in an intelligible fashion and written in standard English?

Reviewer #1: Yes

Reviewer #2: Yes

Reviewer #3: Yes

Reviewer #4: Yes

5. Review Comments to the Author

**Reviewer #1: **The title of the article is very interesting and very meaningful, This is the first comprehensive study of the diverse causes of neuropathy in a territory undergoing epidemiologic transition in the Amazon region. The content of the article is regular and there are no obvious mistakes. It basically wants to express the content clearly and has a certain popular significance

**Reviewer #2:** This article is well written.

Here are my few comments:

1. Research title could be "Pattern,causes and functional outcome of peripheral neuropathies in Amazon region.

2. In lines 93 & 94 AND lines 131 to 134 they wrote about mortality of the general underlying illness like diabetes rather than the topic of interest like diabetic neuropathy,Chemotherapy-related neuropathy ,GBS etc.

3. In line 122-123 they wrote the lost to follow up as 369 which is huge number as compared with the sample size i.e. 754.

4. For Table 1 and figures 1-3 , the title is incompletely written.Also, the tables and figures shall all be put under results section or at the end of the article after the references.

**Reviewer #3:** In this study, Nathalie Deschamps et al. analyze the different peripheral neuropathies present in the Amazon region to elucidate the most frequent etiologies and their correlation with the severity of the neuropathy, based on the Peripheral Neuropathy Disability (PND) functional scores. I believe that Nathalie Deschamps et al. not only perform a very important and necessary analysis, but also that the outcomes of this study could help elucidate the causes leading to most severe symptoms and mortality, and whose study and treatment should thus be prioritized. In addition, the high amount of patients included will help obtaining very robust and reliable results. On the other hand, considering the high amount of patients and the availability of a good statistical method (STATA 18 software), the collection of more information from the patients and the performance of a more thorough analysis would very much improve this manuscript, which at the moment is rather poor. I also believe that this manuscript would benefit from a language revision.

Below please find my comments, which I hope will help improving this study:

- In this study, the authors analyze the correlation between the different etiologies leading to a peripheral neuropathy and their severity. To do this, they use the Peripheral Neuropathy Disability (PND) functional scores. The study would benefit from an analysis of also the pain and sensations related to these neuropathies, with scores such as the Numerical Rating Score (NRS), the Neuropathic Pain Symptom Inventory (NPSI) or the Modified Glasgow Composite Pain Scale (GCPS), for instance. In addition, the authors could use other severity and neuropathy scores, such as the Toronto Clinical Neuropathy Score (TCNS), to validate their results. Since this might not be possible due to lack of information from all patients, this point should be discussed as part of the limitations of the study.

- Through the manuscript, the authors comment on the follow-up of the patients, stating that “Patients without hospital visits by 2023 were considered lost to follow-up.”, that this is the reason for the “unlabeled” group, and that “369 patients were lost to follow-up”. Nevertheless, these 369 are included in Table 1, thus the authors analyze their diagnoses and PND. Why then are the 369 patients “lost”? From where are they excluded? What is the difference between them and the excluded patients due to “insufficient medical information regarding neuropathy characteristics”? In addition, 24 patients were included as unlabeled “because the patients were lost to follow-up.”. Why were these 24 patients not “lost”, like the other 369? The term follow-up means that there would be two time points of analysis, where at the first timepoint there would be 754 patients, and at the second timepoints there would be 385 (because 369 would be lost). Since this study does not include different timepoints, the “lost to follow-up” statement and the inclusion of those 369 patients nonetheless is very confusing. Please resolve this issue.

- In Methods, line 63, the authors state the keywords that they used to find the patients used in the study (“…using keywords such as "neuropathy," "Guillain-Barre syndrome," "chronic inflammatory demyelinating polyneuropathy (CIDP)," "multineuritis," "amyloid," and "amyloidosis".”). In figure 1, however, they show a flow chart of the 754 patients included, where they state that they use the keyword “No neuropathy” to 3746 patients. Please clarify the selection process and change whichever is wrong accordingly.

- In addition, the selection of different neuropathies ("Guillain-Barre syndrome," "chronic inflammatory demyelinating polyneuropathy (CIDP)," "multineuritis," "amyloid," and "amyloidosis") as keywords in the selection of patients creates a bias in the study and in the probability of finding these neuropathies among the total cohort of patients. Please discuss this bias as one of the limitations of the study.

- In Methods, line 73 to 85, “Determining the cause of the neuropathy”, the authors explain the different causes that they studied and how they are categorized ("Guillain-Barre syndrome" (GBS), "alcoholic," "deficiency," "infectious," "chronic renal failure (CRF)," "connective tissue disease (CTD)," "vasculitis," "rheumatism," "hereditary pressure hypersensitivity neuropathy (HNPP)," "CIDP," "FAP," "monoclonal gammopathy," "toxic," or "unlabeled"). In Table 1 and Figure 2, the authors include one cause of neuropathy that has not been discussed or explained in the manuscript: “intensive care”. Please explain the neuropathies present in the two patients that conform this small group.

- In Methods, line 99, the authors state “Assessment of functional severity within each group was conducted through cross-tabulation and Chi-squared tests. Subsequently, ordinal multiple logistic regression was employed to calculate adjusted odds ratios (aOR) compared to diabetes.”. Why were the odds ratios compared to diabetes in particular?

- In Results, the authors analyze the relation between mortality and the different diagnoses, but they do not include this information in Table 1 or in the figures. Thus I cannot judge whether this analysis and its conclusions are correct. Please include that information in the table.

- In Table 1, authors do a good job analyzing the different PND and diagnoses, but in the text, they do not comment on many of the results that they find. In particular, only two sentences correlate the severity (PND) with the diagnoses (line 129-131), and do not give concrete numbers. In addition, the study of PND is of the most important analyses in this study, yet authors do not mention or discuss these results in the Discussion.

- In the Discussion, line 177, the authors state “We may have missed cases from more isolated regions where access to healthcare is limited, potentially leading to an underrepresentation of deficiency-related peripheral neuropathy in our findings.”. Please explain why limited access to healthcare would affect the deficiency-related neuropathies in particular, but not the other ones.

**Reviewer #4:** This is an interesting study reporting on the frequency of neuropathy and its severity in French Guiana. However, the main concern is how the authors define neuropathy.

The authors simply interpreted the clinical data as diagnostic of neuropathy, including paresthesia, sensorimotor deficits and abolition of reflexes (symmetrical? distal? Please clarify). Also, it is possible misdiagnosis since neuropathy was evaluated by internal medicine, infectious diseases, or intensive care departments (line 77). There are no nerve conduction studies, therefore subclinical neuropathy could be underestimated and misdiagnosis could also exist.

In addition, I would like the authors to discuss the following points:

79. Please, clarify what types of deficiency.

86. Include Peripheral Neuropathy Disability (PND) functional scores in the article.

124. Why do you think toxicity is the second frequent cause of neuropathy? It is not common. Please argue this.

125. Guillain-Barré syndrome is the third cause with 48 cases, could you determine any associated infection?

127. What is “Bita”? Is it a regional plant?

Table 1. It is striking that there are no cases of thyroid disorders, that FAP and HNNP occur more than CMT, and that alcohol consumption is not among the main causes.

86. PND functional scores were determined retrospectively from medical records to quantify functional severity at disease onset and last hospital visit. In Table 1, do the PND score data refer to baseline or last visit? It would be interesting to determine how the PND score evolved over time between the first and last visit.

6. PLOS authors have the option to publish the peer review history of their article (what does this mean?). If published, this will include your full peer review and any attached files.

Reviewer #1: No

Reviewer #2: No

Reviewer #3: No

Reviewer #4: No

---

## [Author Response · Author response to Decision Letter 0]

8 Nov 2024

Dear Editor, we are very grateful for the precious reviewer comments. The positive comments were very encouraging and the suggestions have helped us greatly in revising and hopefully improving our manuscript.

Response to reviewers

Reviewer #1: 

The title of the article is very interesting and very meaningful, This is the first comprehensive study of the diverse causes of neuropathy in a territory undergoing epidemiologic transition in the Amazon region. The content of the article is regular and there are no obvious mistakes. It basically wants to express the content clearly and has a certain popular significance

Reviewer #2: This article is well written.

Here are my few comments:

1. Research title could be "Pattern,causes and functional outcome of peripheral neuropathies in Amazon region.

Answer:

Thank you for the suggestion we have changed the title

2. In lines 93 & 94 AND lines 131 to 134 they wrote about mortality of the general underlying illness like diabetes rather than the topic of interest like diabetic neuropathy,Chemotherapy-related neuropathy ,GBS etc.

Answer:

OK, we added “Mortality of patient with PN”

3. In line 122-123 they wrote the lost to follow up as 369 which is huge number as compared with the sample size i.e. 754.

Answer:

Indeed, in the context of widespread poverty, transport difficulties, geographical isolation, intense migration within South America, and low health professional density, follow-up interruptions are a major challenge for all health problems in French Guiana. What the status of these patients is varies, many patients eventually come back, but some never do and it is not clear whether they are still alive or not. In the first version of the manuscript, the 369 patients were considered lost to follow-up when the data were collected in 2023 (line 88) in order to calculate the mortality rate. We performed survival analysis and censored observations after the last visit. We added the following sentence in the limitations section “Finally, in the local context of widespread poverty, transport difficulties, geographical isolation, intense migration within South America, and low health professional density, irregular follow-up is a major challenge for all health problems in French Guiana, which makes the ascertainment of the vital status at a given time uncertain, until they eventually come back. We therefore censored data after the last known status.” 

We also presented the data differently: The different analysis times were added at 1 year, 2 years, 3 years to measure mortality. 

4. For Table 1 and figures 1-3 , the title is incompletely written.Also, the tables and figures shall all be put under results section or at the end of the article after the references.

Answer:

Ok the titles were modified:

Fig 1. Flow chart of 754 patients included

Table 1. Demographic and characteristics of 754 patients included with peripheral neuropathy

Fig 2. Repartition of PND score by causes of peripheral neuropathy

Fig. 3. Repartition of infectious causes of peripheral neuropathy

Fig.4. Association between causes of peripheral neuropathy and severe PND score excluding GBS.

Table 2: Risk factors associated with severe PND at 1 year 

Fig.5 Mean PND score with standard deviation by causes of peripheral neuropathy at baseline, 1 year and 2 years.

Table 3: Causes associated with neuropathic pain

Fig.6 Risk of death stratified according to causes of peripheral neuropathy adjusted for age and sex.

Table 4: Causes of peripheral neuropathy associated with mortality at 5 years

S1 Table. Number at risk and number censored in Kaplan-Meir survey analysis. 

The tables have been placed in the results after the paragraphs where they are cited for the first time. Figure legends have been placed in the results after the paragraphs in which they are first quoted, as required by the journal.

Reviewer #3: In this study, Nathalie Deschamps et al. analyze the different peripheral neuropathies present in the Amazon region to elucidate the most frequent etiologies and their correlation with the severity of the neuropathy, based on the Peripheral Neuropathy Disability (PND) functional scores. I believe that Nathalie Deschamps et al. not only perform a very important and necessary analysis, but also that the outcomes of this study could help elucidate the causes leading to most severe symptoms and mortality, and whose study and treatment should thus be prioritized. In addition, the high amount of patients included will help obtaining very robust and reliable results. On the other hand, considering the high amount of patients and the availability of a good statistical method (STATA 18 software), the collection of more information from the patients and the performance of a more thorough analysis would very much improve this manuscript, which at the moment is rather poor. I also believe that this manuscript would benefit from a language revision.

Below please find my comments, which I hope will help improving this study:

- In this study, the authors analyze the correlation between the different etiologies leading to a peripheral neuropathy and their severity. To do this, they use the Peripheral Neuropathy Disability (PND) functional scores. The study would benefit from an analysis of also the pain and sensations related to these neuropathies, with scores such as the Numerical Rating Score (NRS), the Neuropathic Pain Symptom Inventory (NPSI) or the Modified Glasgow Composite Pain Scale (GCPS), for instance. In addition, the authors could use other severity and neuropathy scores, such as the Toronto Clinical Neuropathy Score (TCNS), to validate their results. Since this might not be possible due to lack of information from all patients, this point should be discussed as part of the limitations of the study.

Answer:

Thank you for the suggestion

A variable “neuropathic pain” was added and defined as “as painful sensation of burning, tingling, numbness, itching, electric discharge or cold.”

The use of questionnaires and other scores were unfortunately not possible due to the incompleteness of the data;

We added the following to the discussion:

“Furthermore, due to the retrospective nature of this study, physical examination data were not systematically performed and were thus not exhaustive, making it impossible to use other scores to characterize PN. In the case of neuropathic pain, there is probably a measurement bias due to the absence of a screening questionnaire which may have led to underestimation.”

- Through the manuscript, the authors comment on the follow-up of the patients, stating that “Patients without hospital visits by 2023 were considered lost to follow-up.”, that this is the reason for the “unlabeled” group, and that “369 patients were lost to follow-up”. Nevertheless, these 369 are included in Table 1, thus the authors analyze their diagnoses and PND. Why then are the 369 patients “lost”?

Answer:

Thank you indeed this needs further explanation. The 369 patients were the patients not seen in 2023 for whom we could thus not ascertain their vital status in order to calculate the mortality rate of neuropathies. A number of patients eventually come back long after their expected consultation but we censored observations after the last news. Perhaps lost to follow-up is a bit strong for has not yet reconsulted yet.

To get a more detailed view we measured mortality at 1 year, 2 years and 5 years. And for the PND we looked at 1 and 2 years. 

Unlabeled patients were patients with a negative etiological assessment.Some patients with a suspected genetic cause or incomplete etiologic assessment were classified as unlabeled because they had been lost to follow-up.

As discussed above, in the limitations we added the following “Finally, in the local context of widespread poverty, transport difficulties, geographical isolation, intense migration within South America, and low health professional density, irregular follow-up is a major challenge for all health problems in French Guiana, which makes the ascertainment of the vital status at a given time uncertain, until they eventually come back. We therefore censored data after the last known status.” 

 From where are they excluded? 

Answer:

Ok we specified the number of patients at each analysis time

What is the difference between them and the excluded patients due to “insufficient medical information regarding neuropathy characteristics”?

Answer:

Unlabeled patients had an evaluation by a neurologist confirming the presence of peripheral neuropathy. The available data made it possible to know the characteristics of the patients, the onset of neuropathy and the determination of the PND score.

Patients were excluded if this information was not available and did not allow calculation of the PND score. 

We added:

“while exclusion criteria encompassed tunnel syndrome, radiculopathies, pre-existing symptoms predating January 2015, and insufficient medical information regarding neuropathy characteristics permitting to calculate the Peripheral Neuropathy Disability (PND) functional scores”

“Patients were classified as “unlabeled” if the results of etiological screening were not available.”

 In addition, 24 patients were included as unlabeled “because the patients were lost to follow-up.”. Why were these 24 patients not “lost”, like the other 369? 

Answer:

Indeed it is confusing we have tried to rephrase and explain. 

Correction

“Patients were classified as “unlabeled” if the results of etiological screening were negative or not available. Also, some patients with a suspected genetic cause or incomplete etiologic assessment were classified as unlabeled because they had interrupted follow-up.”

The term follow-up means that there would be two time points of analysis, where at the first timepoint there would be 754 patients, and at the second timepoints there would be 385 (because 369 would be lost). Since this study does not include different timepoints, the “lost to follow-up” statement and the inclusion of those 369 patients nonetheless is very confusing. Please resolve this issue.

Thank you we have tried to explain more clearly.

We have added analysis time to assess PND evolution at 1 and 2 years

In addition measurement of mortality was performed at 1 year, 2 years and 3 years.

Deletion of “369 lost to follow up” because previously defined as patient not seen during the year 2023 (but many may come back later) in order to calculate the mortality rate at that time.

- In Methods, line 63, the authors state the keywords that they used to find the patients used in the study (“…using keywords such as "neuropathy," "Guillain-Barre syndrome," "chronic inflammatory demyelinating polyneuropathy (CIDP)," "multineuritis," "amyloid," and "amyloidosis".”). In figure 1, however, they show a flow chart of the 754 patients included, where they state that they use the keyword “No neuropathy” to 3746 patients. Please clarify the selection process and change whichever is wrong accordingly.

Answer:

Thank you for pointing this out. In a first phase we captured cases including the single term neuropathy but after reading the extracts containing the key words, in many cases neuropathy was preceded by “absence of” so we excluded these patients who in fact did not have neuropathy. We simplified the flow chart to avoid this confusion. 

Correction

“No neuropathy” appeared in the excerpt from the document featuring the keyword

n=4071

- In addition, the selection of different neuropathies ("Guillain-Barre syndrome," "chronic inflammatory demyelinating polyneuropathy (CIDP)," "multineuritis," "amyloid," and "amyloidosis") as keywords in the selection of patients creates a bias in the study and in the probability of finding these neuropathies among the total cohort of patients. Please discuss this bias as one of the limitations of the study.

Answer:

Correction

OK thank you we added “The use of keywords as “Guillain-Barré syndrome”, “Chronic inflammatory demyelinating polyneuropathy (CIDP), “amyloid” and “amyloidosis” may have led to a selection bias, with an over-representation of patients with these causes of neuropathy.”

- In Methods, line 73 to 85, “Determining the cause of the neuropathy”, the authors explain the different causes that they studied and how they are categorized ("Guillain-Barre syndrome" (GBS), "alcoholic," "deficiency," "infectious," "chronic renal failure (CRF)," "connective tissue disease (CTD)," "vasculitis," "rheumatism," "hereditary pressure hypersensitivity neuropathy (HNPP)," "CIDP," "FAP," "monoclonal gammopathy," "toxic," or "unlabeled"). In Table 1 and Figure 2, the authors include one cause of neuropathy that has not been discussed or explained in the manuscript: “intensive care”. Please explain the neuropathies present in the two patients that conform this small group.

Answer

Correction 

Ok we added the MESH term “Critical illness polyneuropathy” to the methods

- In Methods, line 99, the authors state “Assessment of functional severity within each group was conducted through cross-tabulation and Chi-squared tests. Subsequently, ordinal multiple logistic regression was employed to calculate adjusted odds ratios (aOR) compared to diabetes.”. Why were the odds ratios compared to diabetes in particular?

Answer

Thank you for your vigilance this was a mistake

Patients had an analysis adjusted for age category and gender.

- In Results, the authors analyze the relation between mortality and the different diagnoses, but they do not include this information in Table 1 or in the figures. Thus I cannot judge whether this analysis and its conclusions are correct. Please include that information in the table.

Answer

We added kaplan meier survival analysis in results fig 6 and mortality rate in Table 4. We hope it is clearer.

- In Table 1, authors do a good job analyzing the different PND and diagnoses, but in the text, they do not comment on many of the results that they find. In particular, only two sentences correlate the severity (PND) with the diagnoses (line 129-131), and do not give concrete numbers.

Answer

OK we expanded the discussion. 

“However, after excluding GBS, which is a known severe neuropathy, infectious causes were significantly associated with a severe PND score (aOR=3.69 (1.18-11.58)) (shown in Fig.4). At 1 year, GBS and deficiency were significantly associated with a severe PND score, while only toxic causes remained significantly less severe (Table 2). 

Diabetes showed functional worsening, with a significant increase in PND score at 1 and 2 years. Conversely, toxic neuropathy and Guillain-Barré syndrome improved their functional scores at 1 and 2 years (Fig.5).”

 In addition, the study of PND is of the most important analyses in this study, yet authors do not mention or discuss these results in the Discussion.

Answer

Ok thank you weadded:

“Our study shows that in our region, when faced with severe PN, the first cause to be considered is Guillain-Barré syndrome. However, once this has been ruled out, an infectious cause should be sought, particularly HIV infection or leprosy (shown in Fig. 4). In addition, any neuropathy associated with neuropathic pain should be investigated for alcoholism, vasculitis or amyloidosis (shown in Table 3). Despite treatment, patients with GBS or deficiency neuropathy remain significantly functionally more severe at 1 year (shown in Table 2).

Although diabetes patients have a significantly lower PND score, there is a progressive worsening of function at 1 year and 2 years (shown in Fig.5), highlighting a lack of management of this PN, even though it is detected early (as part of the follow-up assessment of diabetic patients).”

- In the Discussion, line 177, the authors state “We may have missed cases from more isolated regions where access to healthcare is limited, potentially leading to an underrepresentation of deficiency-related peripheral neuropathy in our findings.”. Please explain why limited access to healthcare would affect the deficiency-related neuropathies in particular, but not the other ones.

Answer

You are correct there is no obvious reason we modified as follows. “underrepresentation of some causes of peripheral neuropathies

---

## [Decision Letter · Decision Letter 1]

2 Dec 2024

Pattern, causes and functional outcome of peripheral neuropathies in the Amazon region

PONE-D-24-27124R1

Dear Dr. Nathalie DESCHAMPS,

We’re pleased to inform you that your manuscript has been judged scientifically suitable for publication and will be formally accepted for publication once it meets all outstanding technical requirements.

Kind regards,

Claudia Brogna

Academic Editor

PLOS ONE

Reviewers' comments:

Reviewer's Responses to Questions

**Comments to the Author**

1. If the authors have adequately addressed your comments raised in a previous round of review and you feel that this manuscript is now acceptable for publication, you may indicate that here to bypass the “Comments to the Author” section, enter your conflict of interest statement in the “Confidential to Editor” section, and submit your "Accept" recommendation.

Reviewer #2: All comments have been addressed

Reviewer #3: All comments have been addressed

2. Is the manuscript technically sound, and do the data support the conclusions?

Reviewer #2: (No Response)

Reviewer #3: Yes

3. Has the statistical analysis been performed appropriately and rigorously? 

Reviewer #2: (No Response)

Reviewer #3: Yes

4. Have the authors made all data underlying the findings in their manuscript fully available?

Reviewer #2: (No Response)

Reviewer #3: Yes

5. Is the manuscript presented in an intelligible fashion and written in standard English?

Reviewer #2: (No Response)

Reviewer #3: Yes

6. Review Comments to the Author

Reviewer #2: (No Response)

Reviewer #3: Thank you for taking into account all of our corrections, comments and suggestions. I believe the manuscript is now ready for publication.

7. PLOS authors have the option to publish the peer review history of their article (what does this mean?). If published, this will include your full peer review and any attached files.

Reviewer #2: No

Reviewer #3: No

---

## [Editor Report · Acceptance letter]

9 Dec 2024

PONE-D-24-27124R1 

PLOS ONE

Dear Dr. DESCHAMPS, 

I'm pleased to inform you that your manuscript has been deemed suitable for publication in PLOS ONE. Congratulations! Your manuscript is now being handed over to our production team.

Kind regards, 

on behalf of

Dr. Claudia Brogna 

Academic Editor

PLOS ONE